# Patient Characteristics Associated with Growth of Patient-Derived Tumor Implants in Mice (Patient-Derived Xenografts)

**DOI:** 10.3390/cancers15225402

**Published:** 2023-11-14

**Authors:** Tatiana Hernández Guerrero, Natalia Baños, Laura del Puerto Nevado, Ignacio Mahillo-Fernandez, Bernard Doger De-Speville, Emiliano Calvo, Michael Wick, Jesús García-Foncillas, Victor Moreno

**Affiliations:** 1START Barcelona—HM Nou Delfos, Avinguda de Vallcarca 151, 28023 Barcelona, Spain; 2START Madrid—Fundación Jimenez Díaz University Hospital, Avenida Reyes Católicos 2, 28040 Madrid, Spainimahillo@fjd.es (I.M.-F.); bernard.doger@startmadrid.com (B.D.D.-S.); jgfoncillas@quironsalud.es (J.G.-F.); victor.moreno@startmadrid.com (V.M.); 3Translational Oncology Division, IIS-Fundación Jiménez Díaz-UAM, 28040 Madrid, Spain; lpuerto@quironsalud.es; 4START Madrid—CIOCC HM Sanchinarro, C. de Oña, 10, 28050 Madrid, Spain; emiliano.calvo@startmadrid.com; 5XENOStart START San Antonio, 4383 Medical Dr, San Antonio, TX 78229, USA; michael.wick@xenostart.com

**Keywords:** patient-derived xenografts, PDX, mice, models, translational, prediction, engraftment, tumor growth, cancer, oncology, preclinical

## Abstract

**Simple Summary:**

Successful establishment of patient-derived xenografts (PDXs) requires a profound understanding of the involved factors influencing engraftment success. Little work has focused on analyzing baseline patient’s characteristics that associate with a better outcome in the PDX development process. In a PDX development program of an Oncology Center, data from 585 tumor models are fully analyzed and characterized with this purpose.

**Abstract:**

**Background**: patient-derived xenografts (PDXs) have defined the field of translational cancer research in recent years, becoming one of the most-used tools in early drug development. The process of establishing cancer models in mice has turned out to be challenging, since little research focuses on evaluating which factors impact engraftment success. We sought to determine the clinical, pathological, or molecular factors which may predict better engraftment rates in PDXs. **Methods**: between March 2017 and January 2021, tumor samples obtained from patients with primary or metastatic cancer were implanted into athymic nude mice. A full comprehensive evaluation of baseline factors associated with the patients and patients’ tumors was performed, with the goal of potentially identifying predictive markers of engraftment. We focused on clinical (patient factors) pathological (patients’ tumor samples) and molecular (patients’ tumor samples) characteristics, analyzed either by immunohistochemistry (IHC) or next-generation sequencing (NGS), which were associated with the likelihood of final engraftment, as well as with tumor growth rates in xenografts. **Results**: a total of 585 tumor samples were collected and implanted. Twenty-one failed to engraft, due to lack of malignant cells. Of 564 tumor-positive samples, 187 (33.2%) grew at time of analysis. The study was able to find correlation and predictive value for engraftment for the following: the use of systemic antibiotics by the patient within 2 weeks of sampling (38.1% (72/189) antibiotics- group vs. 30.7% (115/375) no-antibiotics) (*p* = 0.048), and the administration of systemic steroids to the patients within 2 weeks of sampling (41.5% (34/48) steroids vs. 31.7% (153/329), no-steroids) (*p* = 0.049). Regarding patient’s baseline tests, we found certain markers could help predict final engraftment success: for lactate dehydrogenase (LDH) levels, 34.1% (140/411) of tumors derived from patients with baseline blood LDH levels above the upper limit of normality (ULN) achieved growth, against 30.7% (47/153) with normal LDH (*p* = 0.047). Histological tumor characteristics, such as grade of differentiation, were also correlated. Grade 1: 25.4% (47/187), grade 2: 34.8% (65/187) and grade 3: 40.1% (75/187) tumors achieved successful growth (*p* = 0.043), suggesting the higher the grade, the higher the likelihood of success. Similarly, higher ki67 levels were also correlated with better engraftment rates: low (Ki67 < 15%): 8.9% (9/45) achieved growth vs. high (Ki67 ≥ 15%): 31% (35/113) (*p*: 0.002). Other markers of aggressiveness such as the presence of lymphovascular invasion in tumor sample of origin was also predictive: 42.2% (97/230) with lymphovascular vs. 26.9% (90/334) of samples with no invasion (*p* = 0.0001). From the molecular standpoint, mismatch-repair-deficient (MMRd) tumors showed better engraftment rates: 62.1% (18/29) achieved growth vs. 40.8% (75/184) of proficient tumors (*p* = 0.026). A total of 84 PDX were breast models, among which 57.9% (11/19) ER-negative models grew, vs. 15.4% (10/65) of ER-positive models (*p* = 0.0001), also consonant with ER-negative tumors being more aggressive. BRAFmut cancers are more likely to achieve engraftment during the development of PDX models. Lastly, tumor growth rates during first passages can help establish a cutoff point for the decision-making process during PDX development, since the higher the tumor grades, the higher the likelihood of success. **Conclusions**: tumors with higher grade and Ki67 protein expression, lymphovascular and/or perineural invasion, with dMMR and are negative for ER expression have a higher probability of achieving growth in the process of PDX development. The use of steroids and/or antibiotics in the patient prior to sampling can also impact the likelihood of success in PDX development. Lastly, establishing a cutoff point for tumor growth rates could guide the decision-making process during PDX development.

## 1. Background

The use of preclinical models is a core component of translational research in oncology. As one of the first steps in cancer research, it provides the background biological knowledge required for successful drug development and clinical trial designs. Despite the revolution that cancer cell line development has brought for cancer discoveries, it is still challenging to translate the preclinical data into clinical results, and the rate of failure in drug development remains very high [1]. There are several limitations to using conventional cell lines in research, negatively influencing their predictive value regarding activity in cancer types in clinical trials: the diversity of solid tumors with respect to molecular profile and sensitivity to specific drugs; the interpatient variability in drug exposure, difficult to predict from preclinical studies; and the highly variable tumor cell doubling time across different tumor types and within a single tumor. Intratumor heterogeneity is a key factor hampering the utility of in vitro preclinical models. In this setting, there is supportive evidence showing the significant genetic variations existing between a primary tumor and the cell lines deriving from that tumor [2,3], a problem that could be surpassed with the development of patient-derived tumor xenograft models (PDTX or PDX). The models have proven to better predict final clinical responses of cytotoxic drugs [4], leading to an increase in the establishment of disease-specific panels of patient-derived xenografts worldwide [5].

In 1983, Bosma et al. reported the severe combined immunodeficiency (SCID) mutant CB17 mice [6] lacking immune T cells, which made them attractive for human tumor engraftments [7,8]. Further crossing of SCID mice with the non-obese diabetic (NOD) strain led to the development of NOD-SCID mice [9], which lack both T- and B-lymphocytes. Many groups have used NOD-SCID mice for PDX model establishment, with it becoming one of the most used tools for this purpose [10,11,12]. This idea was further developed by the National Cancer Institute in the United States [13], and has since been translated into laboratories worldwide.

In the development of PDXs, many protocols have been proposed from different research groups [13,14]. Researchers have studied and found their own ways to improve PDX engraftment success rate [2,15,16]. The usual protocol implies that pieces of solid tumors are collected from tumor tissues obtained from patients either by surgery or biopsy. These pieces are implanted subcutaneously in mice (subcutaneous transplantation), in the same organ as the original tumors in the patients (orthotopic transplantation), or in the renal capsule in the recipient mouse. Subcutaneous transplantation is not only easier to implant, but also easier to follow for engraftment success [3,4,17].

Several studies have tried to determine predictive factors for engraftment success. Reported success has ranged between 23% and 75%, depending mostly on the tumor type and proliferation rate [18,19,20,21,22,23]. Overall, colorectal (64–89%) and pancreatic (62%) tumors have had high engraftment rates, but low-proliferation tumors such as receptor-positive breast cancers [13,14,15,16,17,18,19,20,21,22,23,24,25,26,27] and neuroendocrine low-grade tumors have shown low success rates, even in the laboratories considered most successful in PDX engraftment in general [24]. Likewise, aggressive and advanced (metastatic) cancers have shown high PDX model success rates compared to less-aggressive and nonmetastatic cancers [18,25,26,27].

### 1.1. Background Findings Associated with Engraftment Success Rate

Amongst the factors so far correlated to a higher engraftment PDX rate, the following are included:

#### 1.1.1. Tumor Stage

Chen Y et al. [28,29,30] found different tumor stages play a vital role in engraftment rate, which can roughly reflect tumor burden In non-small-cell lung cancer (NSCLC), tumor samples from patients with stage II (43/96, 45%) and stage III (25/49, 51%) disease seem to have higher engraftment rates than those from stage I disease (32/145, 22%) [15]. Oh et al. evaluated similar parameters, and concluded tumors arising from advanced diseases would grow better in xenografts, specifically in colorectal cancer. Their results showed xenograft uptake in 4 of 15 (26.7%) stage I tumors, in 41 of 72 (56.9%) stage II tumors, in 50 of 84 (59.5%) stage III tumors, and in 55 of 70 (78.6%) stage IV tumors, with a clear higher uptake rate in more advanced disease [31,32]. Similarly, Jung et al. found that patients’ primary tumor size is a significant factor of the success of PDX models in pancreatic cancer [32,33]. Weroha et al.’s results were also consistent, confirming that tumor stage, tumor grade and presence of ascites correlated with better engraftment rates in ovarian cancer models [34,35]. When it comes to scientific research, choosing samples with high tumor stage may be helpful to establish PDX models.

#### 1.1.2. Sample Origin: Metastases

Researchers have shown metastatic cancers exhibit higher PDX model engraftment rates compared to non-metastatic cancers [36,37,38]. Masanori and colleagues generated a PDX model of human brain metastases of breast cancer in the mouse brain [39]. This method had no perioperative mortality and a 100% (10/10) engraftment rate. In colon cancer PDX research, 100% (8/8) engraftment rate was achieved with samples coming from metastases compared with the 84% (27/32) engraftment rate with primary cancer [39,40]. These data suggest that the capability of tumors to grow serially in mice could be associated with their capability to metastasize and seed distant sites, but no real comparative studies have been performed, and samples are still very small for drawing conclusions [30]. We hypothesize that the probability of engraftment is higher when samples are obtained from metastases relative to primary tumors.

#### 1.1.3. Tumor Type and Subtype

Among various tumors in different studies, breast cancer seems to have the relatively lowest success rate of PDX engraftment, ranging from 21% to 37% [29,30,32,35,39,40,41,42,43]. Since breast cancer is a hormone-dependent disease, hormonal receptor status determines the treatment regimen. Moreover, immunodeficient mice cannot provide the hormones needed for tumor growth after transplantation tissue is engrafted from the human body to mice, which leads to the difficulty in establishing breast cancer PDX models [35]. Thus, transplantation rate for triple-negative breast cancer is relatively higher than other breast cancer types. For example, in triple-negative breast cancer, engraftment has resulted to be significantly higher than HER2 positive and luminal cancers, in line with tumor grade and hormonal environment influencing final engraftment rates [18,26,39,41]. Moreover, the stable take rate of ER-negative (52%) and PR-negative (37%) tumors was noticeably higher than that of ER-positive (2%) and PR-positive (3%) tumors [18,26,40]. On the other hand, colorectal cancer, pancreatic cancer, head and neck cancer and ovarian cancer show acceptable engraftment rates in immunodeficient mice, regardless of subtypes [18,29,32,35,42,44,45]. A large study by Echeverria et al. [18], in which 269 tumor samples were obtained from patients diagnosed with triple-negative breast cancer (TNBC) participating in the ARTEMIS trial, achieved success in establishing 62 models, for an overall intake of 23%. They did not find association between prior therapies and engraftment success [18]. The multivariate analysis for several clinical characteristics demonstrated that lymph node status at diagnosis (all samples were from local or locally advanced disease) did correlate with a higher engraftment rate. Ki67 protein-expression positivity was also correlated with higher success rate, with both results considered statistically significant (*p* = 0.020 and *p* = 0.032, respectively). Other markers such histology, subtype, androgen receptors, ethnicity, race or age were not correlated. Interestingly, and in line with previous findings, tumor tissue samples collected from 308 patients who were diagnosed with non-small-cell lung cancer (NSCLC) which were implanted into immunodeficient mice revealed that squamous cell carcinomas had a higher engraftment rate compared with adenocarcinomas [44]. In GBM, in a study by Sloan et al. [45,46], 69 samples from patients with glioblastoma multiforme were collected for PDX generation, achieving successful engraftment in 37 implanted samples in mice. Interestingly, tumor growth rate was measured between passages, and confirmed to be progressively higher, with 11 samples (15.9%) reaching 40% or more increase in tumor growth rate between the first and third passages [47].

However, sample size in those studies is small, and many other factors remain to be clarified. As part of the START PDX program in Madrid, we sought to collect, develop, and analyze a large set of PDXs with their primary source characteristics, evaluate whether these factors predict success rate, and evaluate further markers for future development.

## 2. Methods

Through the development of a PDX bank, we sought to evaluate which factors from source patients could help predict better outcomes. We conducted a prospective, observational study to evaluate the clinical, pathological, and molecular characteristics that correlate with and predict a higher tumor-growth rate in patient-derived xenografts. A full anonymized dataset included the following clinical variables: age, sex, date of diagnosis, menopausal status, presence of diabetes mellitus and use of metformin, smoking habit, and baseline laboratory parameters (peripheral blood tests considered within 21 days of sampling) such as total lymphocytes, platelets, neutrophils, lactate dehydrogenase levels (LDH); the number of prior lines of therapy the patient received prior to sampling (with regard to tumor extraction for mice implants), the most recent line of therapy administered for the patient´s cancer (within 30 days prior to date of implant), recent use of broad-spectrum antibiotics (either orally or intravenously administered within two weeks of sampling), and recent use of steroids at immunosuppressive doses (prednisone or equivalent over 10 mg/day > 3 days within 21 days of sampling); pathology characteristics of the patient’s tumor such as primary location, histology, grade of differentiation (reported by pathologist as grade 1: well-differentiated, grade 2: moderately differentiated, grade 3: poorly differentiated), presence or absence of lymphatic and neural invasion, and *AJCC 8th edition* TNM status at diagnosis; and molecular characteristics including mutational status when available (KRAS; NRAS; BRAF; EGFR; BRCA). Mismatch repair deficiency (dMMR), microsatellite instability (MSI), estrogen- and progesterone-receptor expression, and/or HER-2 status (when applicable) were also included. Local next-generation sequencing (NGS) data (if available) was collected, focusing on NGS data from the samples finally implanteds. Collected data included sample source (primary tumor or metastases), time to tumor growth (determined from the moment of implant until the moment the tumor implanted achieves a volume of at least 150 mm^3^), and the time to engraftment success (determined from the moment of implant until the final engraftment success moment). Final engraftment success is defined as the moment for the third passage to be performed, which coincides with the final sample collection for storing and banking purposes. Tumor growth rate: tumor growth rate was calculated using the following formula: assuming the tumor growth follows an exponential law, Vt being the tumor volume at time t is equal to Vt = V0 exp(TG.t), where V0 is the volume at baseline, and TG is the growth rate. Approximating the tumor volume (V) by V = 4 π R3/3, where R, the radius of the sphere, is equal to D/2. Consecutively, TG is equal to TG = 3 Log(Dt/D0)/t, Dt being the diameter at time t and D0 the diameter at baseline. To report the tumor growth rate (TGR) results in a clinically meaningful way, TGR can be expressed as a percent increase in tumor volume during one month, using the following transformation: TGR = 100 (exp(TG) − 1), where exp(TG) represents the exponential of TG.

### 2.1. PDX Generation

To create PDXs, fresh tumor samples were collected either from primary tumor or metastases and divided into smaller pieces of 2–3 mm^3^, which were later introduced in Roswell Park Memorial Institute (RPMI) medium with antibiotics. The portions are carefully selected and then submerged in Matrigel^®^ (Corning Life Sciences, New York, NY, United States of America) following standard procotols [48,49]. Subcutaneous implants of human tumor fragments (measuring approximately 3 × 3 × 3 mm^3^) are surgically implanted in the lower back of athymic nude female mice of 5 to 6 weeks of age (species used: Mus musculus, Athymic Nude-Foxn1nu, female, 1.5 months old, 25 g from ENVIGO^®,^ Indiana, IN, United States of America). The tumor was measured weekly and was considered grown when the size of the implanted tumor in mice reached a volume of at least 150 mm^3^. The process consisted of the tumor being passed between mice to allow for further growth when it reached a volume of 1500 mm^3^. It could also have been transferred prior to reaching this size for other reasons, such as poor appearance of the animal (highly suggestive of disease and failure), a reason why all dates by measurements at each timepoint were recorded for analysis. Final engraftment success of the model is achieved for those who accomplish three passages (PX3), since this is the moment when they are sampled, frozen and stored for future research (Figure 1).

Growth in time was evaluated at three different timepoints, and then correlated with the gathered characteristics of the patient´s tumor of origin:(1)PX0: Time elapsed between the moment of patient’s tumor implant in mice, and achievement of tumor growth (reaching a volume of 150 mm^3^). Because we mean to evaluate potential patient and tumor characteristics on the final success, retrieving those models that achieve a minimum initial growth was required.(2)PX1: or *First Passage*: time elapsed between the moment the patient’s tumor implant in mice achieves 150 mm^3^ and the moment the tumor is retrieved from the model and implanted on a second mouse.(3)PX2: or *Second Passage:* time elapsed between the moment of implant in the second mouse, and the moment of implant in the third mouse.(4)PX3: or *Third Passage* (coincides with the definition of final engraftment success).

### 2.2. Statistical Analysis

To achieve the objectives of this study, the clinical characteristics of patients, molecular findings and histological analysis results from tumors excised for xenograft generation were collected and analyzed. A descriptive analysis of the variables was provided according to their measurement scale: frequency (and percentages) for qualitative variables and median (interquartile range, IQR) for quantitative ones. Correlations between quantitative variables were assessed by calculating Pearson’s coefficient. Differences between baseline characteristics were analyzed in relation to success for growth (or engraftment success) as well as tumor growth rate. To establish whether there is a variable trend in tumor growth rates between passages, ROC curves were assessed for each passage, considering its timelines, with the goal of finding a threshold that maximizes Youden’s criteria (the total of the sum of the sensitivity and specificity of the model) for engraftment prediction. Likewise, baseline characteristics were analyzed as potential predictors of TGR and final outcomes.

#### 2.2.1. Assessment of Success or Failure of Engraftment

To determine whether baseline clinical, pathological and molecular characteristics are associated with final engraftment success, a full analysis of the data using the Statistical Package for the Social Science system^®^ (SPSS) Version 25.0 (March 2017) was performed. All variables were qualitative and evaluated for associations using chi-square and Fisher’s exact test. Results are represented with their association following a confidence interval of 95%.

#### 2.2.2. Comparison of Tumor Growth Rates at Each Step

To evaluate whether tumor growth rate at the time prior to the first and/or second passages (PX1 and PX2) could truly influence final engraftment success, comparisons are made with respect to the tumor growth rate in the group achieving the third passage (PX3) and between the PX1 and PX2 groups. In the case of growth rates in the second passage, only the comparison between PX2 and PX3 can be made. Comparisons between groups are made with the Mann–Whitney U test. Median and quartile values have been used instead of means and standard deviations, because the growth rate values follow a very asymmetric distribution, and are therefore very far from the normal distribution. For full information see Appendix A.

#### 2.2.3. Associations of Variables with Tumor Growth Rate in Each Passage

To evaluate the potential association of clinical, pathological, and molecular characteristics considered for the tumor growth rate in the time elapsed between each passage, comparisons are made between the groups with the mentioned variables. Continuous quantitative variables are described with the median and interquartile ranges and are compared using the Kruskal–Wallis test. Qualitative variables are described with frequencies and percentages and are compared using the chi-square test or Fisher’s exact test.

To evaluate the association of quantitative variables, the Spearman correlation coefficient has been calculated together with its *p* value. When this coefficient is statistically significant (*p* < 0.05) it can be interpreted as the following:If the coefficient is positive, the correlation is positive; therefore, the higher the variable, the higher the tumor growth rate should be.If the coefficient is negative, the correlation is negative; therefore, the higher the variable, the lower the tumor growth rate should be.

To assess the association with qualitative variables, medians and interquartile range values were calculated for each category of the variable and were then compared using the Mann–Whitney U or Kruskall–Wallis tests.

## 3. Results

A total of 564 implanted models were evaluated for final analyses. Table 1 and Table 2 represent the general description and distribution of the patients delivering the samples. In summary, baseline characteristics of patients are generally well balanced, except for expected characteristics influenced by general population statistics.

At time of data analysis, of 564 tumor-positive samples implanted into mice, 187 (33.2%) PDXs had achieved successful growth at time of analysis.

### 3.1. Baseline Clinical Characteristics Significantly Correlate with Engraftment Success

The use of antibiotics or steroids prior to implant impacts positively on final engraftment success: the administration of systemic antibiotics was evaluated for all patients that provided samples for implants. Two weeks prior to sampling was considered an acceptable time to find any potential different in engraftments. Analysis showed that 38.1% (72/117) of the samples receiving antibiotics achieved final engraftment vs. 30.7% (115/260) of the group that was not treated with antibiotics. These results were statistically significant (*p* = 0.048), suggesting the use of antibiotics in the last two weeks prior to sampling may enhance success rates (Figure 2A).

Likewise, the administration of systemic steroids in the same period significantly correlated with a higher final engraftment success rate. In this setting, 41.5% (34/48) of samples receiving steroids achieved final engraftment success, against 31.7% (153/329) of the steroid-free samples. These findings were also consistent with pre-specified statistical significance (*p* = 0.048) (Figure 2B).

Post-menopausal status showed a positive correlation with final engraftment success: as reported in the literature, hormonal status in women is associated with the incidence of certain types of cancers, especially those considered hormone sensitive such as breast or endometrial cancers. Interestingly, in our study, menopausal status was found to be associated with final engraftment success: 34.9% (95/272) of samples obtained from postmenopausal patients achieved final collection, vs. 20.4% (10/49) of the samples derived from premenopausal patients (*p* = 0.04) (Figure 2C).

High peripheral lactate dehydrogenase (LDH) levels and prediction of engraftment are considered acute-phase-reactant markers associated with inflammation and cell death, which, when present, indicated poorer prognosis in cancer. Baseline LDH levels were measured in the most recent blood analyses prior to sampling (all within 30 days prior to sampling). High LDH levels were associated with higher engraftment rates. Specifically, of the 187 samples that grew, 74.9% (140/187) had LDH above the upper limit of normality against 25.1% (47/187) with normal LDH levels (*p* = 0.034) (Figure 2D).

Tumor grade and ki67 expression: the higher the better. Grade of differentiation is defined as a measure of cell anaplasia in the sampled tumor and is based on the resemblance of the tumor to the tissue of origin. In our work, we based grading on a three-tier scheme provided by the pathologist evaluating the sample, where three categories are defined: low grade (grade 1, or well-differentiated), intermediate grade (or grade 2) and high grade (grade 3, or poorly differentiated tumors). Low-grade tumors had an engraftment rate of 25% (47/187), the intermediate grade had 34.8% (65/187) and the high grade had 40.1% (75/187) (*p* = 0.043,) suggesting that higher differentiation grades are associated with higher engraftment rates (Figure 3A).

Ki67 is a nuclear protein positively associated with cell proliferation and cancer prognosis. Currently, it is widely used as a cell proliferation marker evaluated through immunohistochemistry to assess the proliferative index of cancers. As expected, higher ki67 levels were also correlated with better engraftment rates: only 8.9% (9/45) of implants derived from patients with primary tumors with low ki67 levels achieved growth, versus 31% (35/113) of engraftment success achieved in the high-ki67 group (Ki67 > 15%) (*p* = 0.002) (Figure 3B).

Other histological characteristics that were associated with higher engraftment rates were the presence of lymphovascular invasion in the tumor sample: 42.2% (97/230) with lymphovascular invasion vs. 26.9% (90/334) of samples with no invasion (*p* = 0.0001), and the presence of neural invasion: 41.8% (59/141) of neural invasion-positive samples achieved growth against 30.3% (128/428) (*p* = 0.008) (Figure 3C,D). Hormone receptor positivity, which has classically been considered not only a prognostic but a positive predictive marker of response to hormonal therapy and chemotherapy in breast and other types of cancer, was also correlated. The expression of hormone receptors (HRs) in several tumor types (84 PDX in total) had an impact on total engraftment rate: 57.9% (11/19) of HR-negative models grew, vs. 15.4% (10/65) of HR-positive models (*p* = 0.0001) (Figure 3E).

Mismatch repair deficiency (MMRd) correlated with higher engraftment rates: mismatch repair deficiency, measured by immunohistochemistry in tumor tissue, describes cells that have mutations in certain genes that are involved in DNA repair (mismatch repair genes: MLH1, MSH2, MSH6 and PMS2). Accumulation of errors lead to unrepaired repetitive DNA sequences, leading to high microsatellite instability. Mismatch repair deficiency leads to a higher antigenic environment, which results in better tumor responses to new immunotherapeutic agents such as checkpoint inhibitors. Mismatch-repair-deficient tumors have been described as lower grade tumor with a better prognosis. In our study, tumors coming from samples with MMRd have a higher engraftment rate: 62.1% (18/29) of MMRd tumors achieved growth vs. 40.8% (75/184) of proficient tumors (*p* = 0.026) (Figure 4). On the other hand, these results are not translated into MSI tumors. Although MSI-high tumors tend to grow more, these results were not found to be clinically significant, possibly due to having insufficient numbers of MSI analyses in contrast with MMRd analyses.

Regarding other baseline characteristics, patients’ demographics such as age or sex, and past medical history such as diabetes and smoking habit did not correlate with final engraftment success. Other analyzed baseline laboratory parameters (peripheral blood tests considered within 21 days of sampling) including total lymphocytes, platelets, neutrophils, and neutrophil/lymphocyte ratios, were also not correlated with final engraftment. Lastly, molecular studies did not show any significant correlation with final engraftment success or failure (Appendix A: Association of Variables with Success in Each Passage).

### 3.2. Tumor Growth Rate Increases throughout Passages, and Can Predict Final Success in Engraftment of Pdx Models

The following table (Table 3) summarizes the comparisons made with respect to the tumor growth rate between passages. Growth rates are described by the median and the lower and upper quartiles (25% and 75% percentiles, respectively). Results show that, in fact, the models achieving final engraftment success have a higher tumor growth rate during the first and second passage than those that fail: median TGR (mTGR) = 317 (CI 95%: 125–546) vs. mTGR = 14.6 (CI 95% −68.2–274) for those only achieving the first passage, and mTGR 64.7 (CI 95%: −1.70–256) for those only achieving the second passage. These results were statistically significant: *p* < 0.002 in both cases (Figure 5).

Likewise, tumor growth achieved in the first and the second passages can help predict final engraftment. Using ROC curves to find the threshold that maximizes Youden’s criteria (the total of the sum of the sensitivity and specificity of the model), establishes a threshold for engraftment prediction. The curve presents an area of 0.72, meaning the capacity that tumor growth rate during the first passage predicts if a third passage is finally achieved. A threshold of 72.8 was selected, for which the sensitivity is 82.4% and specificity 5.9%. This implies that 85.4% of final engraftment successes have a TGR during the first passage superior to 72.8, and that 5.9% of failures present a TGR during PX1 of below 72.8 (Figure 6).

Apart from the threshold found following Youden’s criteria, there are other multiple thresholds associating different sensitivity and specificity values. The potential thresholds are shown in the Appendix A, adding the positive and negative predicted values of each point.

#### ROC Curves for Tumor Growth Rates in the Second Passage

(For complete table see Appendix A).

### 3.3. Baseline Clinical and Histological Variables Can Also Help Predict Engraftment Success

Among the variables evaluated in relation to tumor growth rate for each passage, high LDH levels (>ULN) resulted in a positive coefficient with statistical significance (*p* = 0.009). Interestingly, BRAF mutation status also correlates with higher tumor growth rates in PDX models (the presence indicates a higher TGR) *p* = 0.04, and the presence of neural invasion in histopathological sampling (the presence meaning a higher TGR) *p* = 0.014. These results were particularly evident for PX3 (Appendix A).

## 4. Discussion

Here we report the successful generation of 187 PDX models obtained from 564 tumor-positive samples from patients with different tumor types, for an overall success rate of 33,2% at time of analysis. Overall, tumors showing more aggressiveness had a higher tendency for final engraftment success. More specifically, from the histology standpoint, Ki67% protein expression positivity in tumor tissue, tumor grade of differentiation, and the presence of lymphovascular and neural invasion in sample of origin were statistically correlated with a higher engraftment success rate. All these markers have previously been associated with bad prognosis in different types of cancers [47,48,49,50,51,52]. Ki67% protein expression, a marker measured in most kinds of cancers and considered decisive for therapy in some, like breast cancer, has previously been correlated with a higher engraftment success rates, and has been used in a model for prediction of PDX establishment in the previous literature [53,54]. In our work, most samples were obtained from patients with high Ki67 protein expression levels. Of the 158 samples where Ki67% protein expression was available, 113 samples (20%) had a Ki67 protein expression of >15%. These samples grew significantly better, supporting the fact that more-aggressive, poorer-prognosis cancers are more likely to generate successful PDX models, as previously described [29,55,56].

Tumor grade of differentiation, which in general refers to how much or how little tumor tissue looks like the normal tissue it came from, is a well-established marker of prognosis across all tumor types. In our study, in contrast with Ki67% protein expression levels, tumor grade was relatively well distributed across all samples. Most (42%) came from moderately differentiated tumors, followed by high-grade (34.8%) and low-grade (23.2%). As in previous work, according to our results, the higher the grade, the higher the likelihood of final engraftment, also supporting the fact that aggressiveness and dedifferentiation of the tumors plays an important role in predicting PDX establishment [35,55,56,57]. Similarly, lymphovascular and neural invasion were also markers of prediction of higher success rates, and baseline LDH-blood measured levels which are easily analyzed with a single blood extraction, were also significantly correlated with engraftment. LDH has not previously been associated with PDX engraftment success, although is an easily measurable marker of aggressiveness. Taken together, baseline clinical and histological characteristics can help predict potential final engraftment when developing PDX models and should be taken into consideration during the process [58,59,60].

Most studies have focused on determining whether prior therapies could influence engraftment success rates [19,21,61]. Some have successfully found a potential association, where samples originating from treatment-naïve patients are more likely to succeed as PDX models [4,26,32,41,62]. However, timing between last-treatment dosing and sampling has not been taken into consideration. The possibility of recent anticancer therapy affecting tumor growth in PDXs was an objective of our work. Evaluating any prior therapy within 21 days of sampling was the selected method, since many chemotherapy schemes are administered every 21 days. Different from our predecessors, we could not find any correlation between treatment administration and final engraftment success, even when considering the timing. Unfortunately, too few samples were obtained from patients that had received therapy within 21 days (4%), a limiting factor for this final analysis.

Regarding tumor types, most samples came from colorectal tumors (128 samples), and we could not find any association between tumor types and engraftment success. Focusing on breast cancer models, it is well-established that low grade luminal subtypes are less likely to succeed than more aggressive forms of cancer [23,63,64,65]. Tumor growth rate can help predict success, especially in low-grade tumors. Using the threshold found using Youden’s criteria, the decision on maintaining a model for longer periods of time could help increase final engraftment rates. Most studies consider failure for engraftment when no growth is achieved after 6 months of implant [18,66,67,68,69]. For our breast cancer models, we prolonged this time to 12 months, and found that some luminal subtypes of breast cancer models could achieve growth after 8–10 months of engraftment. Prolonging the time has been attempted with success before [70], and should always be carried out when trying to generate ER-positive PDX models. We have learned that engraftment is significantly higher in HR-negative-expression cancers (57.9% vs. 15.4%, *p* = 0.0001), supporting previous findings [18,26], but most of these studies kept the timing under 6 months, which is an important confounding factor. Considering other associated factors, especially for breast cancer, menopausal status was found to be associated with final engraftment success: 34.9% (95/272) of samples obtained from postmenopausal patients achieved final collection, vs. 20.4% (10/49) of the samples derived from premenopausal patients (*p* = 0.04).

With the revolution of immunotherapy in cancer, the immune microenvironment has taken a big role in PDX development strategies. Although the mice used in this study for PDX generation are immune-compromised mice, we were interested in learning whether the antigenic nature of the sample used for models could impact final engraftment success rates. For this, we evaluated the mismatch repair protein expression in the samples of origin. It is well-established that MMRd tumors are usually low-grade, less-aggressive forms of cancers. Surprisingly, our study confirms that MMRd tumors grow significantly better than MMRp tumors [71,72]. A reason for this inconsistency may be that MMRd tumors usually lead to T cell exhaustion and an immunosuppressive tumor environment and, when an immune-compromised mouse is inoculated, it is able to grow significantly better. In line with this idea, we analyzed whether antibiotic or steroid administration in the 2 weeks prior to sampling could have an impact on engraftment success. Usually, these factors are not taken into consideration, but the truth is that steroid administration at immunosuppressive doses could influence the tumor microenvironment and antibiotic administration could impact on the sample collected [73,74]. In fact, both antibiotics and steroids (at immunosuppressive doses, >10 mg of prednisone or equivalent daily within 2 weeks prior to sampling), did negatively impact on final engraftment success. In both cases, the immune microenvironment is affected, either by the doses of steroids administered, or by the actual immunosuppression arising from a potential coincidental infection prior to sampling. This is the first study to correlate prior steroid and antibiotic administration with PDX engraftment success rate, and more studies should consider broadening our knowledge in this area, since it could lead to an improvement in the processes of PDX modelling. Importantly for our work, microsatellite instability was not found to be associated with final engraftment success, but the availability of these results was significantly lower and was only available for 58 PDXs.

Regarding other baseline characteristics, patients’ demographics such as age or sex, and past medical history such as diabetes or smoking habit did not correlate with final engraftment success. Other analyzed baseline laboratory parameters (peripheral blood tests considered within 21 days of sampling) including total lymphocytes, platelets, neutrophils, and neutrophil/lymphocyte ratios, were also not correlated with final engraftment.

The investigators then sought to determine whether success in PDX generation could be predicted throughout PDX growth. The importance of this evaluation relies on the decision-making process whilst growing PDX models. Is there any way we can help PDX-generation researchers during the process to decide whether to pursue or terminate a PDX model? For this, we introduced the concept of tumor growth rate (TGR—already used in oncology for cancer-response evaluation processes and predictions—REF) into PDX generation. Cancer cells were implanted into mice, and. following a formula, we could build and establish tumor growth rate following a logarithmic, more natural scale for the time between passages. TGR was evaluated during the first, the second and the third passages, with their timelines. Interestingly, tumor cells would replicate better with each passage, meaning TGR progressively increases after each passage for successful models [75], shortening the time for growth after each passage. More importantly, the models that finally achieve engraftment success have a higher TGR during the first passage than those that only achieve the first and/or second passages (*p* < 0.002). The positive impact of confirming this pattern of growth relies on the predictive value we could add to measured TGR for final engraftment success. To measure the potential predictive value, we followed Youden’s criteria and established a TGR threshold of 72.8 during the first passage, which was able to help predict final success with a sensitivity of 82.4%. This threshold could be used during the first passage to help predict which models would finally be successful. We then sought to determine if baseline characteristics could be associated with a higher tumor growth rate in these subsets of patients. Interestingly, the baseline characteristics that associate with a better tumor growth rate are LDH levels in blood prior to sampling, BRAF mutation status in the tumor, and the presence of lymphovascular and perineural invasion in the tumor, at least from a statistical significancy standpoint, suggesting that for these subsets, tumor growth rate plays a particularly important role in the final prediction of success, and that the threshold could be of more value. The work establishes baseline situations that should be considered during the decision-making process of PDX development, and could be of significant impact for researchers given the costs and time invested in these processes. Given the magnitude of importance PDXs have in preclinical and translational research, further investigation becomes necessary to make preclinical ground tool development an easier, more efficient, and more effective task.

## 5. Conclusions

The present research renders further evidence that basal characteristics of patients providing tumor samples for PDX development, and characteristics from the tumor of origin, do have an impact in final engraftment success. Learning about these factors is crucial for ensuring a more efficient PDX development process, and applying the knowledge could have a significant impact in the decision making. Many studies have tried to evaluate these potential predictive markers, but few have been able to establish potential markers for solid tumors, regardless of primary diagnosis. In our work, we show that clinical parameters, such as menopausal status and histological factors such as hormone receptor expression, high grade of differentiation, high Ki67%-protein expression, the presence of lymphovascular or perineural invasion, and molecular alterations such as MMR deficiency are components of the sample of origin that significantly correlate with engraftment success during the PDX development process. Further work is required to assess whether other molecular profiles could help predict engraftment, and using time-associated parameters such as tumor growth rate should help make the process a more efficient one.

## Figures and Tables

**Figure 1 cancers-15-05402-f001:**
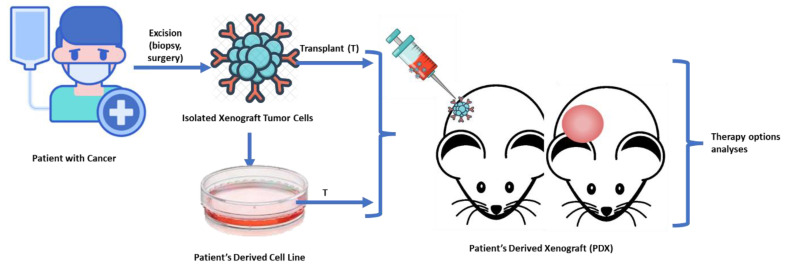
Process of PDX Development: Samples obtained from patients are carefully selected and submerged in Matrigel. Subcutaneous implants of human tumor fragments are surgically implantes in the lower back of athymic nude mice. The implant is progressively measured during its growth.

**Figure 2 cancers-15-05402-f002:**
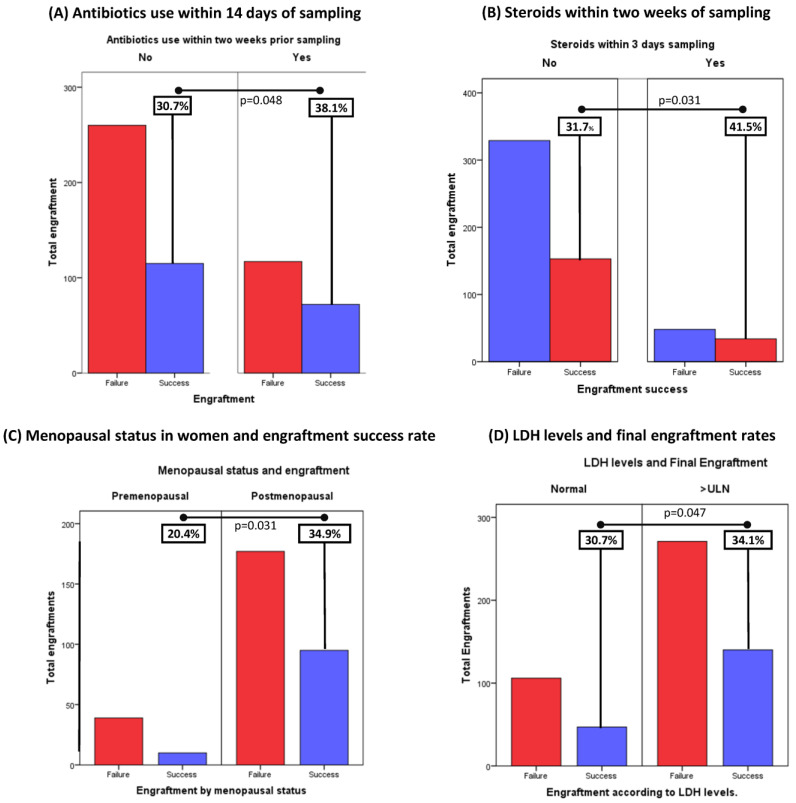
Baseline Clinical Characteristics and their association with final tumor growth in PDXs: (**A**) Analysis showed that 38.1% (72/189) of the samples receiving antibiotics achieved final engraftment vs 30.7% (115/375) in the group that was not treated with antibiotics (*p* = 0.048). (**B**): Interestingly, 41.5% (34/48) of samples receiving steroids achieved final engraftment success, against 31.7% (153/329) of the steroids-free samples. These findings were also statistically significant (*p* = 0.05). (**C**) Menopausal status was found to be associated with final engraftment success: 34.9% (95/177) of samples obtained from postmenopausal patients achieved final collection, Vs 20,4% (10/39) of the samples derived from premenopausal patients (*p* = 0.031). (**D**) High LDH levels were associated with higher engraftment rates. 34.1% (140/411) of samples obtained from patients with baseline LDH levels above the upper limit of normality (ULN) achieved final growth, against 30.7% (47/153) of the samples with normal LDH succeeded (*p* = 0.047).

**Figure 3 cancers-15-05402-f003:**
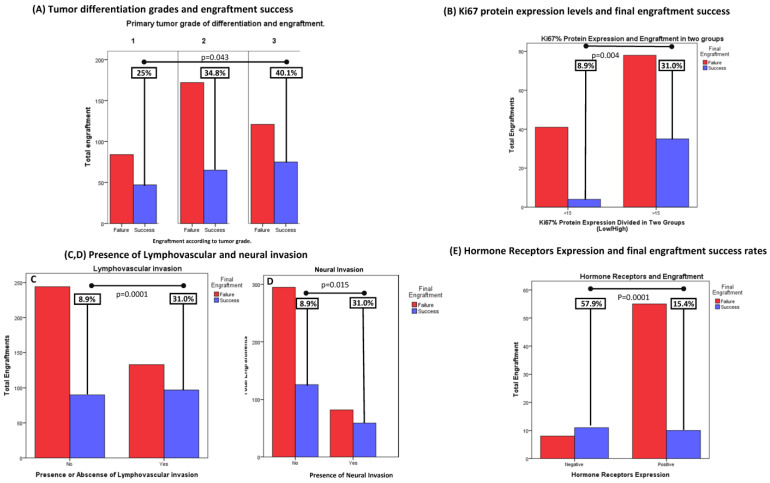
Histological characteristics associated with engraftment success. (**A**) Grade of differentiation reported by pathologist was also assessed. Low grade tumors had an engraftment rate of 25% (47/187), intermediate grade of 34.8% (65/187) and high grade of 40.1% (75/187) suggesting that higher differentiation grades are associated with higher engraftment rates (*p* = 0.043). (**B**) Higher ki67 levels were consistent with better engraftment rates: Only 8.9% (9/45) of implants derived from patients with primary tumors with low ki67 levels achieved growth, versus 31% (35/113) of engraftment success achieved in the high ki67 group (Ki67 > 15%) (*p* = 0.002). (**C**,**D**) Interestingly, 42.2% (97/230) with lymphovascular invasion vs 26.9% (90/334) of samples with no invasion achieved final growth (*p* = 0.0001). Likewise, 41.8% (59/141) of neural invasion-positive samples achieved growth against 30.3% (128/428) (*p* = 0.008). (**E**) The expression of hormone receptors (HR) in several tumor types (84 PDX in total) had an impact in total engraftment rate: 57.9% (11/19) of HR negative models grew, vs. 15.4% (10/65) of HR positive models (*p* = 0.0001).

**Figure 4 cancers-15-05402-f004:**
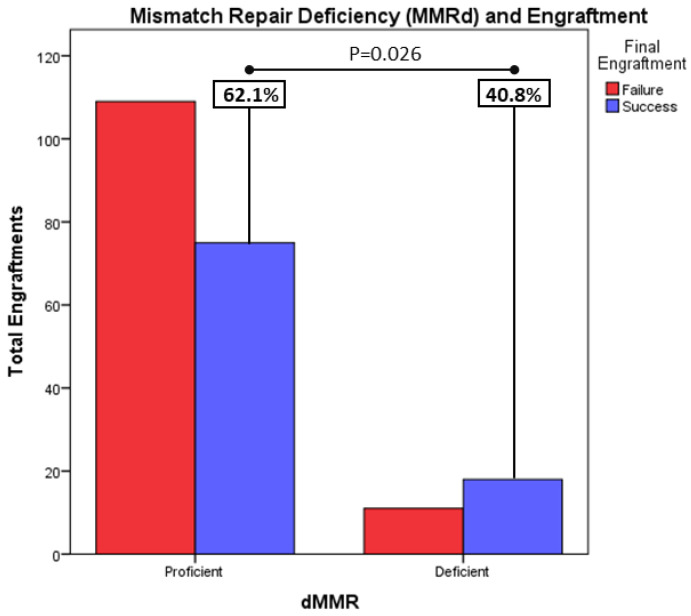
Tumors coming from samples with MMRd, have a higher engraftment rate: 62.1% (18/29) of MMRd tumors achieved growth vs 40.8% (75/184) of proficient tumors (*p* = 0.026).

**Figure 5 cancers-15-05402-f005:**
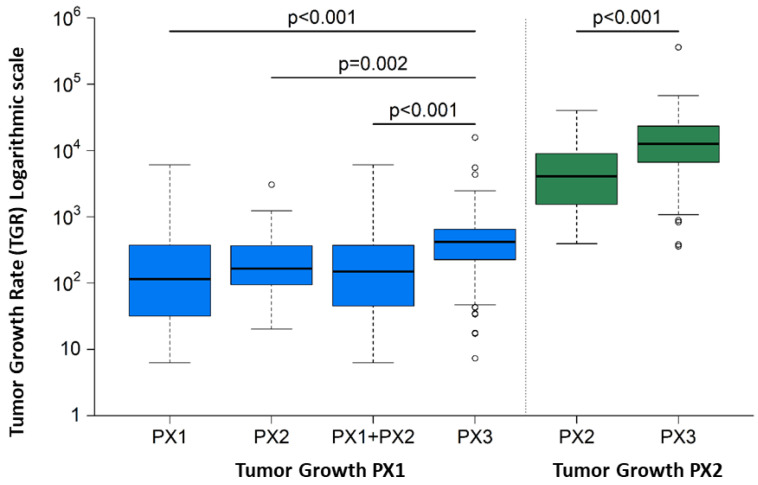
Tumor Growth Rate between models achieving PX1, PX2 and PX3 final success. The models achieving final engraftment success have a higher tumor growth rate during the first and second passage than those that fail: median TGR (mTGR) = 317 (CI 95%: 125–546) vs. mTGR = 14.6 (CI 95% −68.2–274) for those only achieving the first passage, and mTGR 64.7 (CI 95%: −1.70–256) for those only achieving the second passage. These results were statistically significant: *p* < 0.002 in both cases.

**Figure 6 cancers-15-05402-f006:**
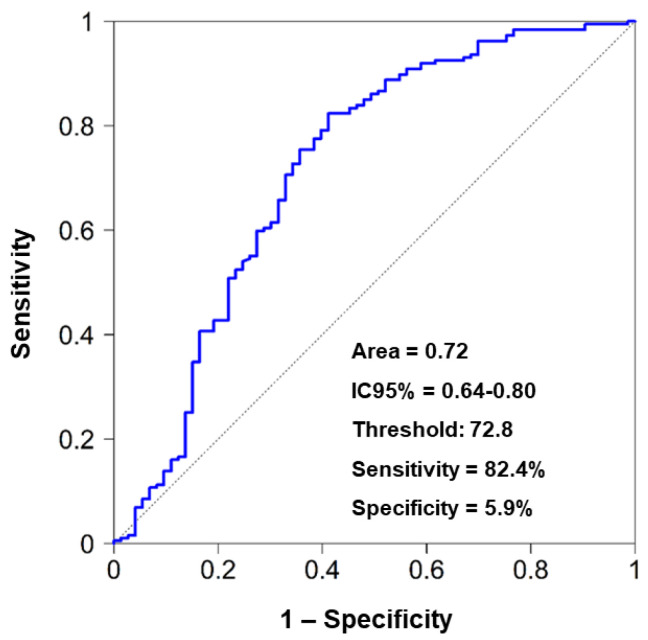
Engraftment success prediction using TGR. Establishing a threshold of 72.8, a sensitivity of 82.4% and a specificity of 5.9% were achieved in predicting engraftment success. This implies that 85.4% of final engraftment successes have a TGR during the first passage superior to 72.8, and that 5.9% of failures present a TGR during PX1 of below 72.8.

**Table 1 cancers-15-05402-t001:** Baseline patient, primary tumor, and sample characteristics.

Baseline Patient Characteristics		*n* (%)
Clinical Characteristics		
Sex	Male	242 (42.9)
	Female	322 (57.1)
Menopausal status	Premenopausal	49 (8.7)
	Postmenopausal	273 (48.2)
Diabetes Mellitus	Diabetic	89 (15.8%)
	Non Diabetic	475 (84.2)
Smoking habit	Non smoker	300
	Smoker	264
Chemotherapy < 21 days ^1^	Yes	23 (4%)
	No	541 (96%)
Primary Tumor Baseline Characteristics		
Tumor grade	High grade (grade 3)	196 (34.8%)
	Middle grade (grade 2)	237 (42%)
	Low grade (grade 1)	131 (23.2%)
Ki67%	<15	45 (8%)
	15–30	22 (3.9%)
	>30	91 (16.1%)
	Total	158 (28%)
MMRd ^2^	Proficient	184 (32.6%)
	Deficient	29 (5.1%)
	Total	213 (37.8%)
HER2 expression	HER 2 Amplified	7 (1.2%)
	HER 2 Negative	81 (14.4%)
	Total:	88 (15,6%)
Hormone Receptor Expression	Positive	62 (11.5%)
	Negative	19 (3.4%)
	Total	84 (14.9%)
BRCAm ^3^	BRCA mutant	17 (3%)
	BRCA wild-type	61 (10.8%)
	Total	78 (13.8)
Sampling characteristics		
Sample source ^4^	Primary	405 (71.8%)
	Metastatic	159 (28.2)

^1^ Refers to chemotherapy administered to patients within 21 days of sampling; ^2^ MMRd: Mismatch Repair Deficiency; ^3^ Presence or absence of BRCA mutation in patient’s tumor; ^4^ Source of initial sample in patients, either from primary tumor or from metastases.

**Table 2 cancers-15-05402-t002:** Implanted tumors and final engraftment (absolute numbers).

Implanted Tumors and Final Growth: Total Numbers	
Tumor Type:	Implanted/Grow *n*
CNS	38/17
Bladder	15/6
Breast	56/11
Luminal Subtype	44/5
HER2 Enriched	4/0
TNBC	8/6
Cervix	12/6
CRC	128/74
Endometrial	28/6
Gastric/Gastroesophageal Junction	11/3
Kidney	49/9
NSCLC	86/20
SCLC	5/3
Ovarian	62/19
Pancreas	8/2
Liver	7/0
Head and neck	9/3
Biliary tract	5/2
Other solid tumors	48/6

**Table 3 cancers-15-05402-t003:** Tumor Growth Rate between models achieving PX1, PX2 and PX3 final success.

Passage	Group	Median (P25%, P75%)	Comparison with PX3	Comparison between PX1 and PX2
**First Passage (PX1)**	PX1	14.6 (−68.2, 274)	<0.001	0.761
	PX2	64.7 (−1.70, 256)	0.002	
	PX1 + PX2	48.3 (−54.8, 270)	<0.001	
	PX3	317 (125, 546)		
**Second Passage (PX2)**	PX2	4079 (1538, 8765)	<0.001	
	PX3	12,514 (6544, 23,214)		

## Data Availability

The datasets analyzed during the current study are available from the corresponding author on reasonable request. The project in self-funded. The PDX Development Program belongs to a collaboration between START San Antonio (XENOStart PDX Development) and the Fundación Jimenez Díaz University Hospital BIOBANCO Program.

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
