# Peer review of "Patient Characteristics Associated with Growth of Patient-Derived Tumor Implants in Mice (Patient-Derived Xenografts)"

_cancers, 2023, doi:10.3390/cancers15225402_

Round 1

Reviewer 1 Report

Comments and Suggestions for Authors

The authors attempt to conduct a research study with the aim of deriving more relevant conclusions, but it becomes challenging. They do not explain which tumor types exhibit greater growth or why they exclusively used female mice. The study is difficult to follow, making it unclear how patient characteristics might influence successful engraftment. The final conclusion " Interestingly, histological factors such as grade, Ki67% protein expression, lymphovascular or perineural invasion, and molecular alterations like MMR deficiency or BRAF mutation are components of the sample's origin that can certainly aid in predicting the final engraftment and should be considered when initiating a PDX program" , "It is not necessary to use animals to obtain this conclusion, which can be observed in the patients' medical records." In general, the more aggressive, invasive, and with a higher Ki67 percentage a tumor is, you could assume that could be a good model. 

Author Response

Thanks very much for reviewing the manuscript and for sharing your thoughts and input. It is highly appreciated to receive constructive criticism to improve the way we write, redact and explain the results of our work. We have tried to address the comments and have added the following changes:

  • A table with each tumor type and absolute numbers for growth has been added. When analyzing the data, there was no real association between tumor type and final growth. A reason for this result could be that the samples are highly heterogeneous regarding histological subtypes, and it is difficult to group them only between tumor origin. For example, in the head and neck group, there are squamous cell carcinomas of oral cavity, nasopharynx, hypopharynx, etc. Same happens with lung cancers, where several histologies are engrafted. We would need significantly more samples to achieve statistically significant results in histological subtypes.
  • Assumptions can be tricky in research, but it is completely reasonable what you mention. Although most results are expected, some are not. For example, although higher grade cancers to grow better in PDX models, we did find MMRd deficient tumors (characterized as usually being low grade tumors) also grow more rapidly. Demonstrating that patient’s characteristics do influence PDX models engraftment rates is of high importance, since most PDX modelling companies focus mainly on technical factors, or factors associated only with the tumor to be implanted.

Reviewer 2 Report

Comments and Suggestions for Authors

This study provides a broad analysis of the independent factors (not associated with the method) that may affect the success of human tumor xenografting in the mouse. This technique (named as PDX) has great potential interest for many fields of research, although in many cases it has proven to be too inefficient. In my opinion, the authors have had a great idea in making this extensive compilation, whose observations and conclusions may be of great use to other research groups in tackling the task of PDX.

There are some issues that have concerned me, and I think that the authors have to provide more information:

- Why was tumor location not a variable to be related to PDX success? In my opinion, it should be the first factor to be mentioned and analyzed for the usefulness of the article. The authors have not mentioned this issue until the fourth paragraph of the discussion, explaining that they found no association between tumor type and graft success. But this sentence should be supported by data, at least in the supplementary table.

- The authors should analyze whether the use of one or the other type of mouse influenced the success of PDX, and provide all the relevant data about it.

- How do the authors explain that MMRd correlated with higher engraftment rates and MSI did not? This explanation should be included in the Discussion. In fact, in the Methods, they stated "MMRd and or MSI" (line 108), which they apparently seem to assume as a single clinical variable.

- A table showing the data results and p-values referred to in the text is required. Even if it is in the supplementary material.

- In the Supplementary Tables, "Previous Biologic" seems to be a variable with statistically significant influence, doesn't it? (the text has broken links and this has made it very difficult for me to read).

- The background section could usefully provide more information on the usefulness of PDX.

MINOR COMMENTS

-  In some places, the writing switches to the present tense, instead of keeping the past tense (e.g. lines 128 and 130).

- Line 227: p= 0.05 is not statistically significant. Maybe it was p= 0.049.

Author Response

: Thank you very much for the thorough review of the manuscript. The work behind it has been long and difficult from the data gathering perspective, and your comments are very much appreciated since they are highly detailed and are meant to help make our manuscript better. We certainly believe the information is very useful, and plan to continue gathering info on over 1000 models already implanted in the program, which should help bring more predictive data to our work.

  • This study provides a broad analysis of the independent factors (not associated with the method) that may affect the success of human tumor xenografting in the mouse. This technique (named as PDX) has great potential interest for many fields of research, although in many cases it has proven to be too inefficient. In my opinion, the authors have had a great idea in making this extensive compilation, whose observations and conclusions may be of great use to other research groups in tackling the task of PDX.

There are some issues that have concerned me, and I think that the authors have to provide more information:

- Why was tumor location not a variable to be related to PDX success? In my opinion, it should be the first factor to be mentioned and analyzed for the usefulness of the article. The authors have not mentioned this issue until the fourth paragraph of the discussion, explaining that they found no association between tumor type and graft success. But this sentence should be supported by data, at least in the supplementary table.

- We have added the information regarding tumor types and absolute numbers of final engraftment success. We have grouped them according to organ of origin, since grouping different histology cancers can become challenging. When analyzing the data, we could not find statistically significant correlations between tumor type and engraftment success. This is likely because of the very heterogeneous samples. We have several different histological subtypes of, ie, lung cancers, ovarian cancers, or even head and neck cancers. To draw significant conclusions we may need to have a larger volume of samples, and group them not only by organ of origin but also by histological subtype, and this falls beyond our line of work at present. We have added the supplementary information with all the associated statistics.

- The authors should analyze whether the use of one or the other type of mouse influenced the success of PDX, and provide all the relevant data about it.

Both are athymic mice, which was the reason for using both. We use female only. Unfortunately, making comparisons between mice is currently impossible with the collected data. Both types of mice were used randomly and the final engraftment rates are not separated between types of mice. Our objective was to evaluate the patient’s characteristics, instead of analyzing the technique. Still, this is a very good idea, and since we are still developing the PDX program, we will start collecting this data for future comparisons and potentially a publication on specific technical information such as mice used, techniques of implant, etc.

- How do the authors explain that MMRd correlated with higher engraftment rates and MSI did not? This explanation should be included in the Discussion. In fact, in the Methods, they stated "MMRd and or MSI" (line 108), which they apparently seem to assume as a single clinical variable.

Information for MMRd tumors was available in a significantly larger number of models, whilst the sample for MSI was lower. The lack of sufficient sample may be one of the reasons for not achieving significance in MSI tumors. We have corrected MMRd and/or MSI, since these are not the same, and was not a single variable analyzed in the work, but two different variables.

- A table showing the data results and p-values referred to in the text is required. Even if it is in the supplementary material.

We have added this to the supplementary.

- In the Supplementary Tables, "Previous Biologic" seems to be a variable with statistically significant influence, doesn't it? (the text has broken links and this has made it very difficult for me to read).

Certainly, but unfortunately biologics include several types of compounds, such as VEGFinh, ICI-I, etc, which make it a result not truly useful in the practice.

- The background section could usefully provide more information on the usefulness of PDX.

We have added some information about the usefulness and prior publications in this setting, trying not to make it too long for the reader. Thanks

MINOR COMMENTS

-  In some places, the writing switches to the present tense, instead of keeping the past tense (e.g. lines 128 and 130).

Many thanks for this input, we have reviewed and redacted better.

- Line 227: p= 0.05 is not statistically significant. Maybe it was p= 0.049.

Yes, this was the case. Thank you very much

Round 2

Reviewer 1 Report

Comments and Suggestions for Authors

The authors have implemented the suggested changes and accurately explained what was requested. For this reason, I believe that this article can be published in this journal as it is now.

Reviewer 2 Report

Comments and Suggestions for Authors

Thank you for your reply.